

# Morphological, ultrastructural and molecular variations in susceptible and resistant genotypes of chickpea infected with Botrytis grey mould

Richa Thakur[1], Rajni Devi[2], Milan Kumar Lal[3], Rahul Kumar Tiwari[4], Sucheta Sharma[1] and Ravinder Kumar[4]

[1] Department of Biochemistry, Punjab Agricultural University, Ludhiana, Punjab, India
[2] Department of Microbiology, Punjab Agricultural University, Ludhiana, Punjab, India
[3] Division of Crop Physiology, Biochemistry and Post harvest Technology, ICAR-Central Potato Research Institute, Shimla, Himachal Pradesh, India
[4] Division of Plant Protection, ICAR-Central Potato Research Institute, Shimla, Himachal Pradesh, India

## ABSTRACT

Biotic stress due to fungal infection is detrimental to the growth and development of chickpea. In our study, two chickpea genotypes *viz Cicer pinnatifidum* (resistant) and PBG5 (susceptible) were inoculated with ($1 \times 10^4$ spore mL$^{-1}$) of nectrotrophic fungus *Botrytis cinerea* at seedling stage. These seedlings were evaluated for morphological, ultrastructural, and molecular differences after 3, 5 and 7 days post inoculation (dpi). Visual symptoms were recorded in terms of water-soaked lesions, rotten pods and twigs with fungal colonies. Light and scanning electron microscopy (SEM) revealed the differences in number of stomata, hyphal network and extent of topographical damage in resistant (*C. pinnatifidum*) and susceptible (PBG5) genotypes, which were validated by stomatal index studies done by using fluorescence microscopy in the infection process of *B. cinerea* in leaves of both chickpea genotypes. In case of control (water inoculated) samples, there were differences in PCR analysis done using five primers for screening the genetic variations between two genotypes. The presence of a Botrytis responsive gene (LrWRKY) of size ~300 bp was observed in uninoculated resistant genotype which might have a role in resistance against Botrytis grey mould. The present investigation provides information about the variation in the infection process of *B. cinerea* in two genotypes which can be further exploited to develop robust and effective strategies to manage grey mould disease.

# INTRODUCTION

The adverse disease symptoms have a significant impact on the growth and development of crop plants, resulting in diminished crop yields and harmful crop products (*Mukta & Bart, 2015*; *Luchi, Renaud & Santini, 2020*). Chickpea is a significant source of protein for the vegetarian population of South Asian nations. The low yield in chickpea is due to its

Corresponding authors
Sucheta Sharma,
suchetasharma_pau@pau.edu
Ravinder Kumar,
chauhanravinder97@gmail.com

susceptibility to diverse fungal pathogens that affect the plant from seedling stage to maturity. Botrytis grey mould (BGM), caused by *B. cinerea* is an important biotic constraint to chickpea production with annual yield loss of 50% or more (*Khan, Anwer & Shahid, 2011*; *Soltis et al., 2019*). However, BGM infection can cause up to 100% yield loss under favorable conditions as reported in Gurdaspur belt of Punjab in Northwest India in 2014–15. The fungus infects all of the plant's aerial components, including the leaves, flowers, pods, branches, and stem, with the flowers, pods, and growth tips being particularly susceptible (*Soliman et al., 2015*). The pathogen exists as *B. cinerea* in the anamorph stage and as *Botryotinia fuckeliana* in the teleomorph stage. Heavy and extended winter rains, cloudiness, heavy nighttime dew, excessive irrigation, early planting, and a dense plant canopy all contribute to the crop's susceptibility to this disease.

Due to high levels of host tolerance to BGM, this disease causes a potential threat to chickpea crops worldwide. However, certain accessions of *C. pinnatifidum* possess a high level of resistance that can be exploited by interspecific hybridization with chickpea. The surface of the plant is the first line of protection against pathogens. A thorough comprehension of plant-pathogen interactions is necessary for mitigating their negative effects. Observable disease symptoms suggest a physiological connection between plant infection and its progression in the host plant (*Francisco et al., 2019*). This enables us to study the quantitative alterations in plant by pathogen infestation. Visual methods allow detecting disease symptoms earlier to more thorough detection (*Duba et al., 2019*). Both light and scanning electron microscopic (SEM) techniques are used to analyze cell surfaces inoculated with *B. cinerea* inoculums. SEM technique was utilised to explore the plant pathogen interaction in greater detail. It has advantages over light microscopy because it allows for the imaging of the entire cell surface, taxonomic features, and improved host-fungus interactions. It is a method that requires minimal material to determine particle size, shape, and texture (*Alves & Pozza, 2009*). It has been demonstrated that molecular markers are an effective tool for understanding pathogenesis and the resistance or susceptibility of the host plant to various infections. The creation of biological products such as phenolics, phytoalexins, osmolytes, antioxidant enzymes, degrading enzymes, and pathogenesis-related proteins by resistance genes activated by plant pathogens confer genetic resistance (*El-Fatah et al., 2019*). Molecular studies on plant pathogen interaction lead to the development of long lasting resistant strategies for crop protection. The present study describes the variation in morphological, physiological and molecular characteristics of two different genotypes of chickpea after they were infected with *B. cinerea*. Visual symptoms of disease measure the morphological variations. Light microscopy and SEM approaches demonstrate the anatomical features after infection. Using PCR analysis, we measure the differences in the expression level after *B. cinerea* infection in both the genotypes using WRKY, BIG and BRG primers which have been suggested to play a positive role to provide resistance against BGM infection in the previous studies (*Smith et al., 2014*; *Sham et al., 2015*).

## MATERIALS AND METHODS

Chickpea seeds of wild *Cicer* species *i.e., C. pinnatifidum* accession 188 and PBG5 (*C. arietinum*) were procured from Department of Plant Breeding and Genetics, Punjab Agricultural University, Ludhiana. For fungal inoculation, the growth chamber technique was used to inoculate PBG5 whereas the cut twig method was used in the case of *C. pinnatifidum*.

### Multiplication of pathogen for inoculations

In the present study, fungal isolate 24, race 510 of *B. cinerea* isolated by the adapted method described elsewhere (*Singh & Bhan, 1986*) was used for BGM inoculation. This isolate was maintained on potato dextrose agar (PDA) slants and further multiplication was done on potato dextrose broth at 25 °C. The spores and mycelium were suspended in sterilized water; passed through a sieve to remove agar lumps and blended for 1 min.

### Growth chamber inoculation technique

Polyethylene pots of size 15 cm × 10 cm were filled with sandy-loam soil under glass house conditions and 10 seeds of PBG5 genotype were sown in each pot. After 1 month, the plants were transferred to chambers in growth room, watered and inoculated with a spore suspension of *B. cinerea* ($1 \times 10^4$ spore mL$^{-1}$). These plants were kept in moist chambers for 1, 3, 5, and 7 days separately with 16 h light and 8 h dark periods provided through a fluorescent lamp (24″ × 1.5″, W 20, 32 m/W).

### Cut twig inoculation technique

In this method, the tender shoots of 1-month-old seedlings of *C. pinnatifidum* accession were cut under water in a tray and wrapped in a wet cotton plug. These twigs were then placed into a test tube containing tap water. For inoculation, twigs were sprayed with a spore suspension of *B. cinerea* ($1 \times 10^4$ spore mL$^{-1}$), covered with moist polythene bags, and incubated as per the method described above for the growth chamber inoculation. Water inoculated plants of both the genotypes served as control. Leaf samples were collected at 1, 3, 5 and 7 dpi and subjected to light microscopy and SEM analysis.

### Anatomical studies

Both the genotypes of chickpea affected with BGM along with controls were studied to monitor the anatomical differences by using light microscopy and SEM.

#### *Light microscopic analysis*

Leaf and stem samples of control and inoculated chickpea genotypes *i.e.,* PBG5 (susceptible) and *C. pinnatifidum* (resistant) were collected after 1, 3, 5, and 7 dpi and immediately preserved in Formalin-acetic acid-ethyl alcohol solution (50% ethyl alcohol, glacial acetic acid and 40% formaldehyde in 85:5:10 ratio). Hand sections of leaves and stem were cut and observed under Leica Bright Field Research Microscope fitted with digital camera and computing imaging systems using software NIS Elements F 3.0 at 20X. Also, the stomatal index was calculated using the method of (*Salisbury, 1927*).

### Scanning electron microscopy analysis

Control and fungal inoculated chickpea leaves were processed for SEM analysis according to the method described (*Alves & Pozza, 2009*). Leaves were cut into small segments with razor blade in thin layers of less than 0.5 to 1 cm. These segments were fixed overnight in 1.5 ml of modified Karnovisk's solution (2.5% each of glutaraldehyde and formaldehyde v/v in 0.05 M cacodilato sodium buffer (pH 7.2), 1 mM $CaCl_2$). Samples were washed in buffer, post fixed with 1% osmium tetroxide solution in same buffer for 4 h in laminar flow chamber at room temperature. The specimens were washed thrice in distilled water and dehydrated using an ethanol gradient (25%, 50%, 75%, 90% and 100%), for 10 min in each concentration. Subsequently, the samples were completely dehydrated in vacuum desiccators. The specimens were pasted using adhesive tapes on the surface of stubs covered with aluminum and submitted to the metallization with gold using ion sputter coater (Hitachi model E-1010) equipment and photographed by scanning electron microscope (Model Hitachi S-3400N, Hitachi Co. Ltd, Tokyo, Japan).

## Molecular studies

To see the genetic variations between two genotypes of chickpea so as to explore the basis of resistance against fungal pathogen, PCR assays were conducted. Different primers enlisted in (Table S1) were used to detect the Botrytis responsive gene in chickpea genotypes.

### DNA extraction, quantification and PCR analysis

The experiment was conducted in Pulse section laboratory, Department of Plant Breeding and Genetics, PAU, Ludhiana. In the experiment, the genomic DNA from both the genotypes of chickpea *i.e., C. pinnatifidum* and PBG5 was isolated using standard cetyl trimethyl ammonium bromide (CTAB) procedure by *Doyle & Doyle (1990)*.
The composition of CTAB buffer used is given as CTAB-1.5%, Tris HCL-100 mM (pH 8.0), NaCl-1.4 M, EDTA-20mM (pH 8.0), β-Mercaptoethanol 2%, Polyvinylpyrolidone-2%. The concentration of purity of DNA was checked by 0.8% agarose gel electrophoresis. The final concentration of DNA (15 ng/µL) in 20 µl reaction mixture for each sample was used to run PCR with following components: DNA-2 µl, Buffer-4 µl, $Mg^{2+}$−1.2 µl, DNTPs-3 µl, Primer (F/R)-1.5 µl, Taq polymerase- 0.2 µl, autoclaved $H_2O$-6.6 µl *etc*. The reaction conditions used were: denaturation at 94 °C, 2 min; 98 °C, 10 s, annealing at 58 °C, 30 s, extention at 68 °C, 1 min for 45 cycles, using Thermo Cycler Applied Biosystems (Veriti). The PCR product was loaded on a 1.5% agarose gel and run at 150 V for 2 h. The gel was illuminated under UV light in gel documentation system.

### Electrophoretic studies of proteins

Electrophoretic studies were carried out by following the methodology of *Laemmli (1970)*. Samples of protein (250 µg) were denatured with three fold SDS than the quantity of the protein in the extract, β-mercaptaethanol and bromophenol blue (0.01%) (tracking dye). Marker of protein having molecular weight ranged between 14.4–97.4 kDa were also applied in a separate well along with the test sample. Samples were loaded at room

temperature to carry out electrophoresis at constant current of 1.5 mA per well until dye stacked to the end of slab gel.

## Staining of gels

Gels were kept in fixing solution for 1 h to fix the resolved protein and then allowed to stand overnight for staining in the staining solution. In the next day, gels were placed in destaining solution on shaker. Washing of the gel was done with destaining solution for 2–3 times until the gel became transparent and clear bands appeared.

## Densitometry

Molecular weights and relative front values of different protein bands were estimated by scanning of the gels through the Alpha-Imager software.

# RESULTS

In the present studies, the plant-pathogen interactions between fungal pathogen *B.cinerea* and two genotypes of chickpea varying in susceptibility to BGM disease have been observed by SEM technique for the first time. Differentiation of specialized cell types or infection structures is a prerequisite for successful penetration and colonization of plant tissues by most fungal pathogens. In this report, microscopic studies of chickpea stem infected with *B. cinerea* allowed a detailed description of the sequential development of infection. A destructive rate of infection was observed on PBG5 leaves as compared to *C. pinnatifidum*, which *B. cinerea* failed to infect.

## Morphological studies

One of the responses that plant employ to cope with diverse biotic and abiotic stresses is the variation in their morphology. The morphological transitions may enable the better idea of the progression of the pathogen in the host plant.

### Visual symptoms

Under optimum environmental conditions, disease symptoms on the PBG5 leaves started appearing after about 48 h of inoculation of the crop with maximum disease symptoms observed on 7th dpi. The diseased leaves showed the appearance of spore colonies on the twigs at 3rd dpi followed by water-soaked lesions on the leaf lamina at 5th dpi and infected leaves without pods (Figs. 1A–1E). Any visible damage to the plant was not found in the case of *B. cinerea* inoculated leaves of resistant genotype *C. pinnatifidum* (Figs. 1K–1O).

## Anatomical studies

### Light microscopic analysis

Different plant parts of chickpea genotypes PBG5 (susceptible) and *C. pinnatifidum* (resistant) were studied under a fluorescence microscope to understand the development of *B. cinerea* infection. Anatomical studies of the cross sections of infected and uninfected stem samples revealed that the outer epidermis of non infected samples was intact (Fig. 1F) while the fungal inoculated samples showed ruptured epidermis in both the genotypes (Figs. 1G and 1Q). Transverse sections of non infected samples revealed intact mesophyll while in case of infected samples, mesophyll disintegration was observed. The intact/

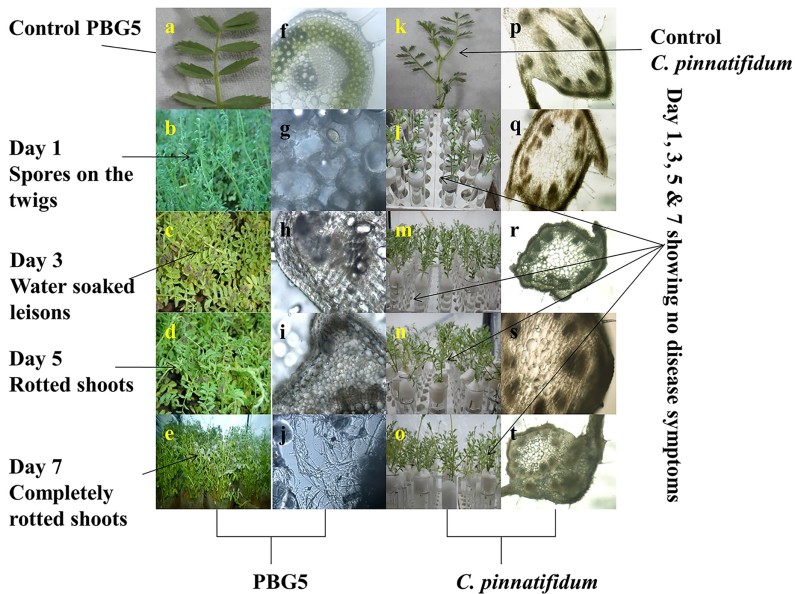

**Figure 1** **Disease symptoms on leaves and light microscopic analysis of stems of *B. cinerea* inoculated chickpea genotypes.** (A) Control PBG 5, (B–E): visual symptoms of BGM infection on susceptible genotype PBG5 at 1, 3, 5 & 7 days post inoculation: (B) greyish colony on the twig, (C) water soaked lesions on the leaf, (D) infected plants without pods, (E) unhealthy plants. (F to J) anatomy of infected PBG five samples at various resolutions by fluorescence microscopy, (F) control PBG 5, (G) day 1 showing colonization of *B. cinerea* conidia in cortex cells, (H) colonization of vascular system, (I) progression of fungal colonization, (J) hyphal network formation, (K) control *C. pinnatifidum*, (L–O): resistant genotype at 1,3, 5 & 7 dpi without any visual disease symptoms, (Q–T): anatomy of inoculated *C. pinnatifidum* showing rigid outer epidermis with progressive days after inoculation.

compact spongy mesophyll in *C. pinnatifidum* could be responsible for hampering mycelium growth in the host tissue and thus imparting resistance against *B. cinerea* infection. This information might play an important role to determine the invasion of the pathogen on host plant. Structural disintegrations were less in resistant chickpea genotype as compared to susceptible one post *Botrytis* inoculation. At 1 dpi, fungal infection affected the average vascular system area of the susceptible genotype, but there were no variations in the number of vessels between susceptible and resistant plants (Figs. 1G and 1Q). At one day post-inoculation, both genotypes exhibited browning of the vascular system. After establishment of infection, mycelia or conidia further proliferated and colonized neighboring areas in the tissue. Another striking feature of the resistant interaction was the observation of thick outer epidermis with the progression of infection. Susceptible genotype PBG5 showed hyphal growth in the cortex of stem and colonization of vascular system at 5 and 7 dpi while in resistant genotype *C. pinnatifidum*, some fungal patches and thickening of outer epidermis was observed after infection (Figs. 1I, 1J, 1S and 1T). The transverse section of leaves of susceptible and resistant genotypes showed difference in the structure of mesophyll cells in the lamina of both genotypes (Figs. 2A and 2B). Also, the numbers of stomata per unit area of leaves of PBG5 were more as compared to resistant genotype (Figs. 3A and 3B, Table S2).

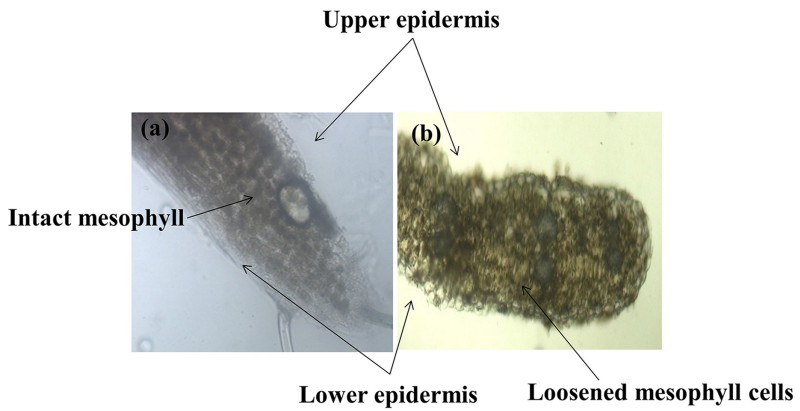

**Figure 2 Anatomy of leaf structure of *C. pinnatifidum* (resistant) and PBG 5 (susceptible) after *B. cinerea* infection.**

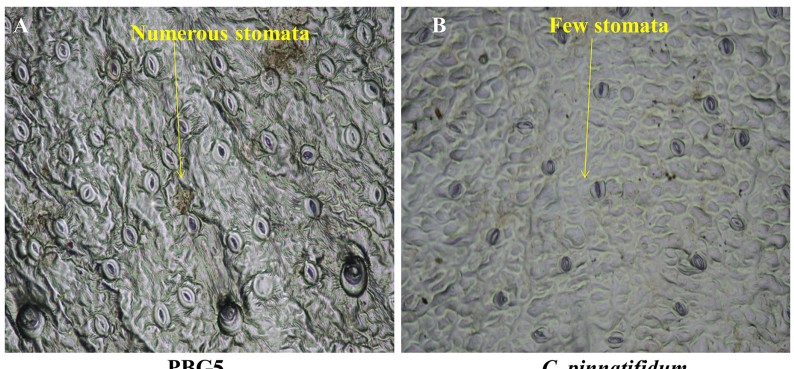

**Figure 3 Stomatal frequency of abaxial surface of leaf sample of susceptible (PBG5) and resistant (*C. pinnatifidum*) chickpea genotypes.**

### SEM analysis

To assess the ultrastructural alterations in control and BGM-infected leaf samples of susceptible and resistant chickpea genotypes, SEM analysis at various resolutions *viz.*, (500 SE, 1.0 k SE, 1.5 k SE, 2.0 k SE and 2.5 k SE; SE stands for secondary electrons detected by detector to provide cell surface topography) were performed (Fig. 4). In the control (uninoculated) samples of both genotypes, leaf surface topography and the number of stomata were seen to vary (Fig. 4, control). The uninoculated leaf samples of the susceptible genotype (PBG5) exhibited higher numbers of stomata than the resistant one, which were also validated by the results of stomatal frequency (the numbers of stomata per unit area) of leaves of both the genotypes. PBG5 leaves have more stomata on the abaxial surface than *C. pinnatifidum leaves* (Fig. 3). The susceptible genotype exhibited a more apparent fungal growth with partial disintegration of cell surface, whereas the resistant genotype exhibited the presence of tiny fungal granules without cell surface damage (Fig. 4, day 1). At 3$^{rd}$ dpi, more disintegration of surface was observed due to fungal invasion in case of susceptible genotype, while minor hyphal formations were seen on the surface of resistant genotype (Fig. 4, day 3). Moreover, the surface structure of susceptible genotype

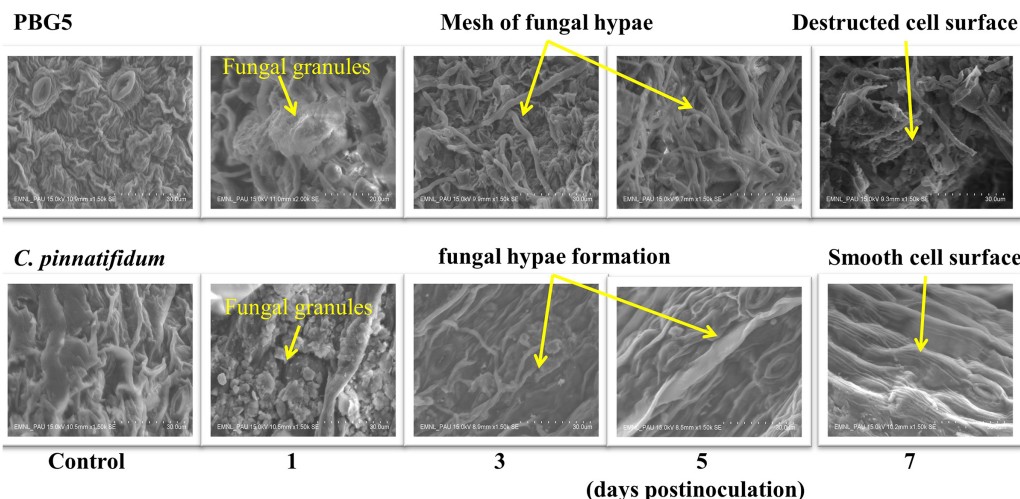

**Figure 4** SEM analysis of uninoculated control and *B. cinerea* inoculated leaf samples of susceptible (PBG5) and resistant (*C. pinnatifidum*) genotype at 1, 3, 5, and 7 days post inoculation.

leaf samples was found to be rough and irregular, whereas the surface structure of resistant genotype leaf samples was smooth and homogeneous. At day 5, hyphal growth development increased in both susceptible and resistant chickpea genotypes as fungal infection increased (Fig. 4, day 5). At day 7 after BGM infection, the entire leaf surface was covered with irregular fungal hyphae in susceptible genotypes and uniform hyphae in resistant genotypes (Fig. 4, day 7).

## Molecular studies

### Detection of Botrytis-responsive genes in chickpea genotypes

Genomic DNA from the uninoculated leaves of both the genotypes was subjected to amplify with five different Botrytis responsive genes specific primers. Out of five, one primer (LrWRKY4) amplified a segment of ~300 bp band in *C. pinnatifidum* only (Fig. 5). Amplification of ~300 bp band was observed in *C. pinnatifidum* but not in PBG5, indicating a possible role of this gene in resistance against BGM (Fig. 6).

## Electrophoretic analysis of chickpea proteins in response to *Botrytis* grey mold

### SDS-PAGE analysis

Total soluble proteins were extracted from uninoculated control and inoculated leaves of chickpea genotypes post *B. cinerea* infection and were subjected to SDS-PAGE analysis. The gels were subjected to densitometry for molecular weight determination and data on $R_f$ values of various protein subunits is presented in Table 1. A total of eight protein bands were detected in uninoculated leaf samples of both the genotypes with molecular weight in the range of 8.0–97.4 kDa (Fig. 7A). Major changes in the polypeptide subunit bands of both the genotypes were observed after inoculation at 3 dpi (Fig. 7B). The inoculated leaves of PBG5 showed six protein subunits bands on gel with $R_f$ values (0.09, 0.16, 0.29, 0.39,
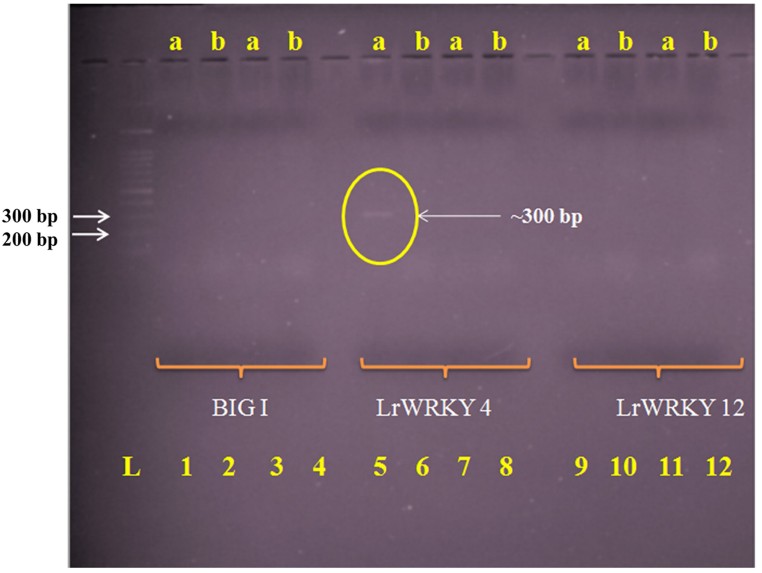

L - marker (1 kb), a - *C. pinnatifidum*, b - PBG5, Lane (5-7 ) - LrWRKY primer

**Figure 5** **Amplification of *C. pinnatifidum* (resistant) and PBG5 (susceptible) genomic DNA with different Botrytis responsive gene specific primers.**

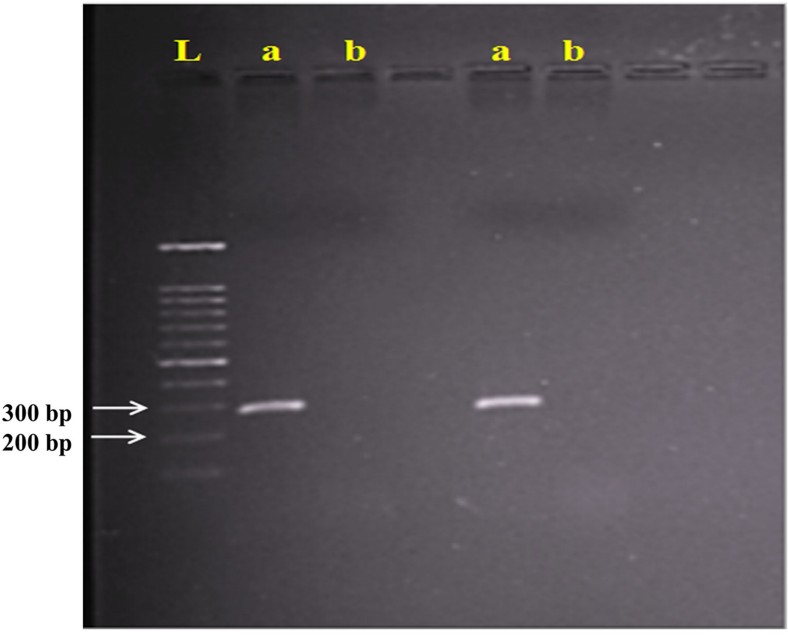

L - marker (1 kb), a - *C. pinnatifidum*, b - PBG5

**Figure 6** **PCR amplification of *C. pinnatifidum* genomic DNA with LrWRKY4 gene specific primer.**

0.57, 0.84) at 3 dpi, (0.09, 0.16, 0.19, 0.28, 0.36, 0.39) at 5 dpi and (0.06, 0.09, 0.14, 0.18, 0.25, 0.36) at 7 dpi. The polypepetide bands I, II, III, IV (M. wt 70.8 kDa, 45.7 kDa, 18.6 kDa, 8.6 kDa & $R_f$ 0.09, 0.16, 0.28, 0.39) were absent in extract from inoculated leaves of

**Table 1** It represents the relative front values and molecular weights of different protein bands estimated by scanning of the gels through the Alpha-Imager software.

| Protein fractions | Days post inoculation | | | | | | | |
|---|---|---|---|---|---|---|---|---|
| | Control | | 3 | | 5 | | 7 | |
| | S | R | S | R | S | R | S | R |
| I | 0.05 (97.4) | 0.05 (97.4) | 0.09 (75.2) | 0.06 (95.5) | 0.09 (75.2) | 0.16 (42.8) | 0.06 (97.0) | 0.15 (43.0) |
| II | 0.11 (66.0) | 0.11 (66.0) | 0.16 (42.8) | 0.11 (66.0) | 0.16 (42.8) | 0.27 (29.2) | 0.09 (75.2) | 0.29 (28.4) |
| III | 0.15 (43.0) | 0.16 (42.8.) | 0.29 (28.4) | 0.16 (42.8) | 0.19 (35.6) | 0.41 (16.9) | 0.14 (42.0) | 0.40 (17.9) |
| IV | 0.21 (39.0) | 0.23 (28.8) | 0.39 (18.0) | 0.21 (39.0) | 0.28 (27.8) | 0.51 (11.7) | 0.18 (34.0) | 0.48 (11.6) |
| V | 0.29 (28.4) | 0.31 (28.2) | 0.57 (11.9) | 0.30 (28.4) | 0.36 (16.1) | 0.60 (8.0) | 0.25 (24.6) | 0.57 (7.9) |
| VI | 0.39 (18.0) | 0.38 (17.6) | 0.84 (7.0) | 0.40 (17.6) | 0.39 (18.0) | 0.75 (6.7) | 0.36 (16.1) | 0.84 (7.0) |
| VII | 0.49 (12.0) | 0.52 (11.5) | | | | | | |
| VIII | 0.60 (8.0) | 0.60 (8.0) | | | | | | |

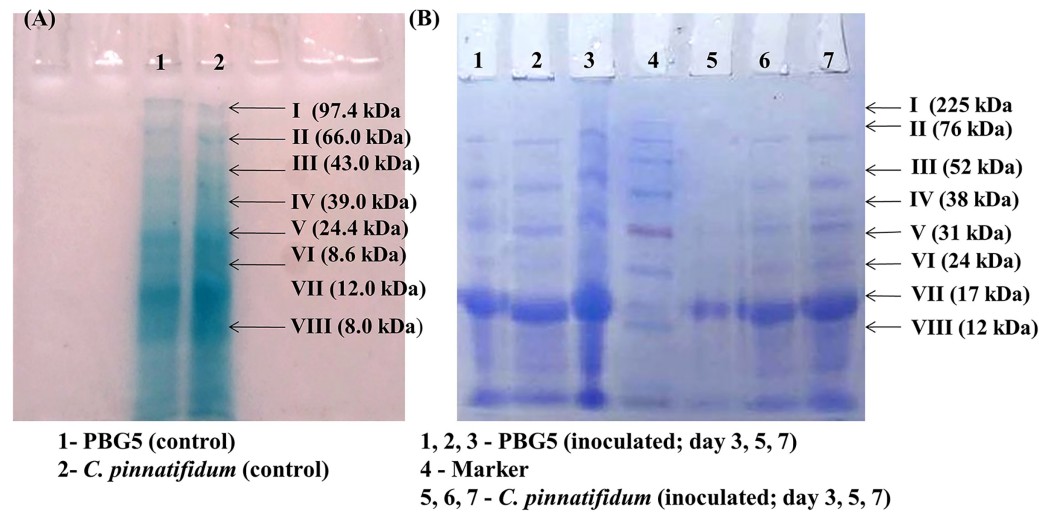

1- PBG5 (control)
2- *C. pinnatifidum* (control)

1, 2, 3 - PBG5 (inoculated; day 3, 5, 7)
4 - Marker
5, 6, 7 - *C. pinnatifidum* (inoculated; day 3, 5, 7)

**Figure 7** Sodium dodecyl sulphate polyacrylamide gel electrophoresis of the leaf extracts of chickpea genotypes PBG5 (susceptible) and *C. pinnatifidum* (resistant) uninoculated and inoculated with *B. cinerea*.

*C. pinnatifidum* at 3 dpi. Protein bands with $R_f$ values (0.5, 0.21, 0.49, 0.64) were not observed in PBG5 leaves with progress of infection at 7 dpi. Inoculated leaves of *C. pinnatifidum* showed absence of protein subunit bands corresponding to $R_f$ value of 0.49, 0.60, 0.64 and 0.84. However, at 3 dpi, the bands corresponding to protein subunits with high molecular weight (97.4 & 66 kDa) were missing in the *C. pinnatifidum* leaves.

## DISCUSSION

A number of biotic stresses exploit metabolic activities of plants due to physiological damage, leading to reduction in crop yield. Plants deploy a well managed and systematic defense mechanism in response to these stresses (*Gimenez, Salinas & Manzano-Agugliaro, 2018*). Thus, a deep understanding of plant defense mechanisms might avoid economic

losses and prevent important crops. A plants first line of defense is passive and involves the acceleration of different physical (waxes, thick cuticles and specialized trichomes) and chemical responses during plant-pathogen interactions (*Nejat & Mantri, 2017*).
The chickpea plant's aerial parts are vulnerable to BGM (*Grewal, Pal & Rewal, 1992*). The susceptible genotype plants that were injected exhibited symptoms comparable to those reported by other writers (*Knights & Siddique, 2002*; *Rahman et al., 2012*). Firstly, it appears as water-soaked lesions on one stem and later on infects other stems. The fungus forms greyish brown to light brown lesions on leaflets, branches, and pods. With the progressive growth of fungus, shoots shed off at the rotting point leading to the formation of rotting mass of the affected leaves and flowers (*Bakr, Rahman & Ahmed, 2002*, *Pande et al., 2002*). Water-soaked symptoms were also reported on detached lily leaves inoculated with *B. elliptica* after 12 h at high humidity and temperature of 20 °C (*Hsieh & Huang, 1998*). Another study conducted by *Ingram & Meister (2012)*, showed visual symptoms of BGM in the leaf, stem, and fruit of tomato. At 4 dpi, two isolates of *B. cinerea*, specifically B191 but not B8403, infected the leaves of tomato plants, as reported by *Rahman et al. (2012)*. In addition, the incidence of disease severity and lesion extension was greater in the B191 isolate than in B8403. *McGrann & Brown (2018)* observed initial small brown pepper spots on the leaves of *Ramulariacollo cygni* inoculated plants of different barley cultivars after 8–10 dpi. The disease spread at a faster rate in susceptible cultivars (Braemar) in comparison to partially resistant cultivar (Power). These spots were converted to large lesions with disease progression ultimately leading to senescence in the susceptible cultivar. Furthermore, another investigation, described the variation in necrotic response in faba bean against *B. fabae* and *B. cinerea* (*Lee et al., 2020*). The cultivar showed aggressive spreading infected with *B. fabae* whereas a generalized invasion was reported toward *B. cinerea*. *Fahrentrapp et al. (2019)*, also recorded *B. cinera* infection in tomato leaves at a pre-visual stage every 80 min using 5-band multispectral sensor. The spectrum observed the reflectance of green, blue, red, near infrared (NIR, 840 nm), and red edge (RE, 720 nm) in time-course experiments of detached tomato leaves inoculated with the fungus *B. cinerea*.

Scanning electron microscopy (SEM) enables the visualization of the whole surface of the cell. It has been widely used electron microscopy technique to estimate particle size, shape, and texture with minimum quantity of sampling material. *Botrytis* species can penetrate the host tissues by hyphae or germ tube infection through epidermal cells, stomata or wounds and this generally involves different mechanical and chemical processes (*Fang Heish, Wen Huang & Hsiang, 2001*). During *Botrytis* infection of the host, there might be mechanical rupture of cuticle due to the presence of tip of germ tube or enzymatic dissolution of epidermal cell walls (*Cole, Dewey & Hawes, 1998*). The presence of small holes with sharp edges in the cuticle of *Vicia faba* leaves suggested chemical degradation by esterase during *B. cinerea* infection (*McKeen, 1974*) whereas *Heuvel & Waterreus, (1983)* observed germ tubes, appressoria and infection of *B. cinerea* on french bean leaves. *Fang Heish, Wen Huang & Hsiang (2001)* used SEM approach to study *B. elliptica* infection on oriental lily leaves and reported that the fungal penetration occurred *via* short swollen germ tube appressoria near stomata on abaxial surfaces of lily

leaves. *Mondal et al. (2013)* recorded a gradual and sequential destruction in the cellular structure of chickpea (*Cicer arietinum* L.) seedlings given higher treatments of cadmium by SEM analysis. Similarly, a unique level of colonization by *streptomyces* species in roots of chickpea was observed by *Gopalakrishnan et al. (2015)*, compared to uninoculated plants, done by SEM. Our results are also in accordance with *Babu et al. (2018)*, who observed ultrastructural changes in native and chemically treated spores of fungus *Curvularia lunata* by SEM analysis. Their observations revealed smooth and uniformly distributed conidiospores in the native preserved form whereas chemically fixed spores showed distorted and extensively shrinked structure.

Significantly higher stomatal size in susceptible watermelon genotype than resistant one in response to *Alternaria* blight (*Mahajan & Dhillon, 2000*) and on grape genotypes against anthracnose disease (*Gurjar et al., 2015*), have been reported earlier. Lesser and smaller stomata in the resistant genotype *C. pinnatifidum* may be implicated in providing resistance by inhibiting pathogen entry, in contrast to PBG5 with a bigger size and a higher stomatal index, which permits pathogen access and suggests their function in *B. cinerea* susceptibility. Variable susceptibility grape cultivars to anthracnose disease exhibited substantial variation in leaf shape and leaf sinus structure (*Murria et al., 2019*). In the present study, the appearance of dense mycelia growth with no mechanical disintegration in the resistant genotype, suggest the possible role of hydrolytic enzymes of the pathogen on chickpea leaves. These observations were also in accordance to the studies in tomato fruit infected by *B. cinerea* (*Diaz, ten Have & van Kan, 2002*) and cut roses infection by *B. cinerea* (*Ha et al. 2021*).

In the present study, the appearance of dense mycelia growth with no mechanical disintegration in the resistant genotype, suggest the possible role of hydrolytic enzymes of the pathogen on chickpea leaves. These observations were also in accordance to the studies in papaya fruit infected by *Colletotrichum gloeosporioides* (*Chau & Alvarez, 1983*) and Barley leaf infection by *Erysiphe graminis* (*Kunoh et al., 1988*).

*Alves & Pozza (2009)* observed the presence of mycelium and oval microconidia on the surface of cotton seeds inoculated with *Fusarium oxysporum*. On the surface of common bean seeds inoculated with *Penicillium* sp., they also noticed larger conidiophores with an elliptical form, as well as globose to subglobose conidia with *Aspergillus flavus*. *Rahman et al. (2010)* observed development of infection hyphae directly from the germinating conidium on the surface of soybean seed within 6 h after inoculation of *Fusarium moniliforme*. Hyphal network and roughed conidiophores of *Drechslera tetramera* were observed on wheat plants whereas smooth walled and slight curves of conidia of *Bipolaris orghicola* were found to be attached to the geniculate conidiophores (*Gulhane, Giri & Khambalkar, 2018*) by SEM analysis. These authors suggested SEM to be an effective technique to observe microconidia and mycelium during fungal infections on different plants as reported earlier (*Rahman et al., 2010*; *Gulhane, Giri & Khambalkar, 2018*).

Multigenic characteristics are accountable for plants' resistance to diverse diseases (*Dean et al., 2012*). To comprehend the relevance of these genes in battling fungal infections, profiling of gene expression must be performed. Such studies have been carried out in various plant species such as *Arabidopsis* and tomato in response to necrotrophic

pathogens (*AbuQamar, Moustafa & Tran, 2016*; *Sham et al., 2014*; *Smith et al., 2014*; *Sham et al., 2015*). The transcription factor named WRKY33 was reported to be responsible for resistance against necrotrophic fungus *B. cinerea*. A number of genes including *BIG*s and *BRG*s, involved in plant responses to stress, metabolism, transport, energy and signal transduction pathways have been induced by *B. cinerea* (*Sham et al., 2017*; *Han, Niu & Liang, 2022*).

*Bezier, Lambert & Baillieul (2002)* reported the expression of a *Botrytis*-responsive gene possessing similar homology to the tobacco hypersensitivity-related hsr203J from grapevine leaves infected with *B. cinerea*. According to the study by *Sham et al. (2017)*, WRKY33 transcription factor was reported to be involved in non-enzymatic pathways of plant stress responses and may be responsible for resistance to the necrotrophic fungus *B. cinerea*. Likewise, 23 *LrWRKY* genes with complete WRKY domains were identified from the *Botrytis*-resistant species *Lilium regale* by *Cui et al. (2019)*. Also, the constitutive expression of these genes in *Arabidopsis* resulted in more resistant plants to *B. cinerea* than wild type plants. For further confirmation studies, the expression of WRKY4 genes in *C. pinnatifidum* needs to be studied after inoculation with *B. cinerea*.

Proteomics represents a remarkable tool to explore protein expression, splice variants, and erroneous or incomplete prediction of gene structures in databases in plant-pathogen interactions. Proteins are considered as the immediate perpetrators of biological actions or responses and have become an indispensable source of genetic information. However, multiple factors such as protein abundance, size, hydrophobicity, and other electrophoretic properties limit the applicability of the proteomic approach at the whole cell level (*Pandey et al., 2006*). Comparative or quantitative proteomic analysis of plants subjected to contrasting stressful treatments is the principally utilized subarea of proteomics to identify responsive proteins that may be involved in mechanisms of stress susceptibility or tolerance. It aims at ascertaining the differences in protein profiles between two samples from different individuals or from distinct treatments (*Jorrin-Novo et al., 2018*). Furthermore, the relatively small numbers of proteomic studies on plant-pathogen interactions have mainly been carried out on bacterial and fungal diseases of foliar tissues (*Quirino et al., 2010*). *Chang et al. (2012)* extracted proteins of ground chickpea seeds by SDS-PAGE and identified α and β subunits of legumin and vicilin subunits. Likewise, SDS-PAGE analysis of lupin seedlings infected with *Fusarium* wilt revealed induction of different PR proteins in susceptible genotype as compared to healthy control (*Mohamed, Abd El-Rahman & Mazen, 2012*). *Jaiswal et al. (2014)* identified the membrane-associated dehydration-responsive membrane proteins involved in signaling such as putative serine/threonine kinase, calcineurin-like phosphoesterase family protein, guanine nucleotide-binding protein subunit, beta-like protein, membrane protein CH1-like and histone acetyltransferase GCN5. Among these, the serine/threonine kinases may act as receptors, which interact with other proteins to affect a wide array of processes, especially in stress adaptation and developmental regulation. In the majority of eukaryotic cells, glycosylation is among the common post-translational modifications. Substantial progress has been made in the previous decade in the context of glycosylation in pathogenic fungi (*Liu, Talbot & Chen, 2021*). The fungal infection in plants was corroborated by the process

of N-glycosylation, O-glycosylation and glycosylphosphatidylinositol (GPI). The functions of these processes in key regulatory mechanisms were linked with appressorium formation, host penetration, biotrophic growth and immune evasion. The diversity of seed proteins using SDS-PAGE technique was studied by *Chittora et al. (2017)*, in three different chickpea genotypes having different seed coat color. The study reported a total of 24 polypeptide bands, out of which 20 were common in all the genotypes and four were polymorphic. Protein profile study of chickpea-wilt interaction by a group of *Kheni et al. (2020)* observed a significant difference in expression in resistance and susceptible genotypes.

## CONCLUSIONS

A number of plant diseases that result in substantial crop losses have been documented on a global scale. For the effective and proper management of any plant disease, an early and accurate diagnosis is required. Thus, the first step in investigating any disease is its prompt detection, which is mostly based on its symptoms. Consequently, monitoring visual symptoms in *Botrytis*-infected chickpea genotypes improved morphological differentiation between two genotypes. Similarly, scanning electron microscopy was a superior method for examining the surface topography of *B. cinerea*-inoculated leaf surfaces of susceptible and resistant chickpea genotypes. SEM enabled to monitor the fungal growth in the form of mycelium at earlier stages and then to a network of hyphae that covered the entire cell surface of leaves inoculated by *B. cinerea*. In addition, the detection of a *Botrytis*-responsive gene in an uninoculated *C. pinnatifidum* genotype provides an additional piece of information that can be used to expedite the development of new resistant varieties against a variety of fungal infections through the use of multidisciplinary approaches. Future molecular studies of these two distinct genotypes following inoculation with fungi could reveal other elements of plant pathogen interactions. This information could be assessed to detect the molecular markers associated with resistance against *B. cinerea*.

### Funding
The authors received no funding for this work.

### Competing Interests
Ravinder Kumar is an Academic Editor for PeerJ.

### Author Contributions
- Richa Thakur conceived and designed the experiments, performed the experiments, analyzed the data, prepared figures and/or tables, and approved the final draft.
- Rajni Devi conceived and designed the experiments, performed the experiments, analyzed the data, prepared figures and/or tables, and approved the final draft.
- Milan Kumar Lal conceived and designed the experiments, authored or reviewed drafts of the article, and approved the final draft.

- Rahul Kumar Tiwari conceived and designed the experiments, authored or reviewed drafts of the article, and approved the final draft.
- Sucheta Sharma conceived and designed the experiments, analyzed the data, prepared figures and/or tables, authored or reviewed drafts of the article, and approved the final draft.
- Ravinder Kumar conceived and designed the experiments, analyzed the data, prepared figures and/or tables, authored or reviewed drafts of the article, and approved the final draft.

## Data Availability

The raw data are in the Supplemental File.

## Supplemental Information

Supplemental information for this article can be found online at http://dx.doi.org/10.7717/peerj.15134#supplemental-information.

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
