# Peer review of "Morphological, ultrastructural and molecular variations in susceptible and resistant genotypes of chickpea infected with Botrytis grey mould"

_PeerJ, doi:10.7717/peerj.15134_

## Round 0.1 · original submission · Major Revisions

Two experts assessed your manuscript and found several flaws that need to be addressed before considering the publication of this study in Peer Journal. The concerns are related to missed results that must be incorporated in the main text and the lack of controls in the PCR reactions and amplicons sequencing. Moreover, the manuscript structure needs to be modified to present logically the methodology, results, and discussion. English usage should also be improved.

Reviewer 1 ·

Basic reporting

The authors present interesting work on morphological and molecular change in two genotypes of chickpea infected by B. cinerea. They demonstrated differences between resistant (C. pinnatifidum) and susceptible (PBG5) genotypes in number of stomata, hyphal network and LrWRKY gene expression only in the resistant plant. However, the manuscript contains some technical issues that need to be resolved consideration for publication. Below are more detailed comments.

Mayor Comments
Some data are missing in the manuscripts as example line 158 “PBG5 leaves was more destructive”. What meaning more destructive? Are authors have data to confirm this asseveration?. Similar case when the authors said (Line 184) “greater number of trichomes” or Line 202 “exhibited higher numbers of stomata than the resistant”, but in both cases there is not table, graph or others quantifiable data to confirm this asseveration.
The figure 1 need to improve, the plant pictures are too small, it is difficult to see the symptoms. On the other hands, the figure should include the days and incorporate an arrow to indicate what need to point out in the pictures from F to J.
Unfortunately, the paper's discussion sometimes seems to be an introduction with data and a summary of the obtained results rather than a proper discussion. The references used in the manuscript some of them are too old.

I think English of the manuscript should be improved. The use of words and structure of some sentences are not easy to understand. I recommend the use of English editing service.

Minor Comments
1. Line 13. What mean C. pinnatifidum?
2. Line 34 the sentence “ low yield in chickpea is due to its susceptibility to diverse fungal…”. It is difficult to believe that as unique reason why there is low yield in chickpea is only due fungal infection.
3. Line 35. What mean BGM?
4. Line 41 Botryotinia fuckeliana should be in cursive.
5. Line 45 What mean disease poses?
6. Line 54-58. The sentence “To explore the plant pathogen interaction….” , it seems to be material and methods, but not suitable for an introduction
7. Line 73 “inoculation,the”, there is not space between comma and the following article or word and should be separate. Similar case is found in all the manuscript.
8. Line 79. “weresuspended”, the two words should be separate. Similar case is found in all the manuscript. Example line 86, 90, 91, 112 among others.
9. Line 133-134. Why did the authors use negative temperature for PCR reaction?
10. Line 174. “fluorescence microscopeto”. Please explain what equipment has been used.
11. Line 180. What mean “forhampering”?
12. Line 194. Figure 2 should indicate Botrytis cells by an arrow.
13. Line 198. What mean “SEM analysis at various resolutions was performed”?. ¨Please explain.
14. Line 206. Figure 4, please indicate by arrow what need to point out in the pictures and incorporate the days in the left of the figure.
15. Line 214-247. These paragraphs should be in discussion.
16. Line 252. There is not information in the introduction why the authors chose the BIG 1,
LrWRKY12, LrWRKY4, W4, and W12 genes for this study.
17. Line 263. Figure 7 is not easy to see the missing band.

Experimental design

no comment

Validity of the findings

The conclusion does not highlight the principal findings of the manuscript.

Reviewer 2 ·

Basic reporting

No comment

Experimental design

It is unclear if method used for artificial inoculation was the same for both plant genotytpes (lines 89-91)
Previous research on grey mould resistance in Cicer spp. is missing in the cited literature. The references in the manuscript should be checked.

Validity of the findings

Method used for molecular analysis is not convincing: are the PCR primer pairs used appropriate for both the genotypes? How do you exclude the presence of possible PCR inhibitors in DNA extract that did not produce amplicons? The amplicon should be sequenced and changes in gene expression after pathogen challenge investigated in the resistant genotype by qPCR.
Protein content in the infected samples is affected by the pathogen presence. This should be discussed in the paper.

---

## Round 0.2 · Major Revisions

I thank the authors for the manuscript's improvement. However, after assessment by peers, the manuscript still is faulty in both format and content. There is an expression analysis requested by the reviewer that I think complements well the results already included in the manuscript.

Reviewer 1 ·

Basic reporting

The authors have improved the ms, however there is some technical issue should be improved before publication.

Experimental design

no comment

Validity of the findings

The figure 1 is not easy to understand, the figure should include the days and incorporate an arrow to indicate Botrytis infection. The legend of the figure 1, I think there is a mistake, in the line 8 say “ (i-o): resistant genotype at 1,3 ,5 & 7 dpi without any”, maybe is (l-o)?, otherwise, I cannot understand the figure.
Unfortunately, the paper's discussion sometimes seems to be an introduction with data and a summary of the obtained results rather than a proper discussion. From line 250 to 281, the authors only describe the diseases and symptoms like water-soaked, but I cannot see a real discussion of results. In the line 300 say “in contrast to PBG5 with a bigger size and a higher stomatal index”, however there is not table or data to support this asseveration, a one photography (figure 3) is not enough to me. Line 313 the authors say “role of hydrolytic enzymes of the pathogen on chickpea leaves “, however they support this affirmation with very old references, in the literature there are plenty of references about the role of hydrolytic enzymes in B. cinerea during infection.
Depth analysis is request about LrWRKY gene expression, there is evidence that LrWRKY genes expression could be in response to defense-related hormones salicylic acid (SA), methyl jasmonate (MeJA) and hydrogen peroxide, however in the ms did not mentioned. What´s mean the LrWRKY 4 is expressed in PBG5 in compared to resistant genotype? How many LrWRKY gene group there are?.
The conclusion does not highlight the principal findings of the manuscript.

Minor Comments
Figure 3 Missing space in the word “C. pinnatifidum” below the photography B, same thing in Fig 4.
Figure 7B please replace the letter “k” by “kDa” and change their position like figure 7A.
Line 23. The word Botrytis is written in minuscule and cursive, sometime in minuscule but not cursive. Please check in the ms and use only one format.
Line 115. Please check the word “with1%”
Line 136. The colon should be close to the word “were”
Line 184. A dot is necessary in the word “B cinerea”
Line 197. Please provide data to support the following sentence “numbers of stomata per unit area of leaves of PBG5 were more as compared to resistant genotype”.
Line 237. Check the word “kDa&”
Line 260. Please check the space in the word B. cinerea
Line 271. Please check word “cinera”
Line 274-325-331-333. The word Botrytis cinerea should be B. cinerea

Additional comments

no comment

---

## Round 0.3 · accepted · Accept

The authors addressed all the concerns previously raised. As a consequence, the manuscript is suitable for publication.

Reviewer 1 ·

Basic reporting

no comment

Experimental design

no comment

Validity of the findings

no comment